# Management of Antracnosis with Electrochemically Activated Salt Solutions (EASSs) on Bean Culture

**DOI:** 10.3390/biology12070964

**Published:** 2023-07-06

**Authors:** María Guadalupe Marquez-Blanco, Yunny Meas-Vong, Brenda Z. Guerrero-Aguilar, Samantha de J. Rivero-Montejo, Luis Miguel Contreras-Medina, Irineo Torres-Pacheco, Ramón Gerardo Guevara-González, Mario Martin González-Chavira, Adrian Esteban Ortega-Torres

**Affiliations:** 1Grupo de Ingeniería en Biosistemas, Centro de Investigaciones Aplicadas en Biosistemas (CARB-CIAB), Facultad de Ingeniería, Universidad Autónoma de Querétaro, Santiago de Querétaro 76010, Mexico; pipitamqz99@gmail.com (M.G.M.-B.); rivermon2014@gmail.com (S.d.J.R.-M.); miguel.contreras@uaq.mx (L.M.C.-M.); irineo.torres@uaq.mx (I.T.-P.); ramonggg66@gmail.com (R.G.G.-G.); 2Centro de Investigación y Desarrollo Tecnológico en Electroquímica (CIDETEQ), Parque Tecnológico, Pedro Escobedo 76703, Mexico; yunnymeas@cideteq.mx; 3Instituto Nacional de Investigaciones Forestales, Agrícolas y Pecuarias (INIFAP), Campo Experimental Bajío, Celaya 38496, Mexico; guerrero.brenda@inifap.gob.mx (B.Z.G.-A.); gonzalez.mario@inifap.gob.mx (M.M.G.-C.)

**Keywords:** legumes, phytophatogenic fungi, plant elicitation, plant biostimulants, eco-friendly agrochemical

## Abstract

**Simple Summary:**

Electrochemically activated salt solutions (EASSs) are widely used as surface disinfecting agents with low cost and high efficiency, because they are a safe option for those having contact with the solution. Following this logic, the solutions may be useful for inhibiting pathogens affecting plants in agricultural systems. Some tests have been made in plant species such as tobacco and apple trees, where they have worked in activating plant natural defenses but also increasing plants’ growth rate. These results suggest an elicitor effect of EASSs on bean plants. In this work, we evaluated this trait by e antioxidant and enzymatic activity assays in addition to determining the inhibitory effect of EASSs on multiple microorganisms with agricultural importance. EASS treatment as an agricultural practice offers several advantages to the current production systems to develop more economically, environmentally, and healthy sustainable crop technologies.

**Abstract:**

Common bean (*Phaseolus vulgaris* L.) is an important crop for food security and for national economics for several countries worldwide. One of the most important factors of risk in common bean production is the fungal disease anthracnose caused by *Colletotrichum lindemuthianum*, which, in some cases, causes complete yield losses; this kind of plant disease is usually managed through the application of chemical products such as fungicides that are commonly not accepted by society. This rejection is based on the relationship of pesticides with health damage and environmental contamination. In order to help in solving these drawbacks, the present work proposes the use of electrochemically activated salt solutions (EASSs) as a safer pathogen control agent in crops, due to it having shown an elicitor and biostimulant effect on plants. With this background, this manuscript presents in vitro results of the evaluation of the inhibitory effect for multiple bean pathogens and in vivo results of EASS in the common bean–*Colletotrichum* pathosystem by evaluation of the infection severity and defense activation, such as secondary metabolite production and antioxidant activity. EASS presence in growth media had a strong inhibitory effect at the beginning of experiments for some of the evaluated fungi. EASSs showed an effect against the development of the disease when applied in specific doses to prevent distress in plants.

## 1. Introduction

Common bean (*Phaseolus vulgaris* L.) is a grain legume that is considered a nutraceutical food worldwide [1], mainly due to its high content of protein [2], anthocyanin, polyphenols, and flavonoids [3]. In addition, the high diversity of common bean varieties allows both low- and high-income regions to produce it, being, for many countries, an important crop for food security and national economics [4,5]. Additionally, the common bean represents 85% of the world’s production of legumes, providing benefits in the form of food to about 300 million people from its cultivation [6].

The necessary production of common bean has an increased risk of loss due to abiotic and biotic variables. One of the most important biotic variables of risk is fungal diseases that infect the crop, such as anthracnose caused by *Colletotrichum lindemuthianum* [7], that, once infected, has been reported to yield a loss of 95 to 100% [8]. The fungus overwinters in seed and crop residues before infecting all bean plants, reaching complete yield losses for susceptible beans [9]. The most distinctive symptoms of anthracnose are manifested in bean pods as deep and shrunken lesions containing visual flesh-colored spores [1]. The symptoms appear in developing and full-grown plant tissues, frequently forming necrotic areas and finally leading to withering, wilting, and death of infected plant tissues [10].

Currently, the main approach to control fungal diseases in crops is the application of chemical pesticides due to their high efficacy and low cost, but there is an increasing negative perception from the population that has associated the excessive use of chemical products in productive systems with severe health and environmental issues [11,12]. Researchers have been focusing on the development of new technologies for agriculture application that allow producers to reduce the use of pesticides in their crops while maintaining (or increasing) the quality and quantity of production [13,14].

Electrochemically Activated Salt Solutions (EASSs) are saline solutions that, due to an electrolysis activating process, contain oxidant agents and are widely reported as disinfectants and cleaners used in the food industry. They are highly effective, economical, organic, and non-corrosive to the human epithelium, as reviewed by [15]. EASSs have shown in vitro antimicrobial activity for different pathogens even higher than that of commercially available disinfectants [16,17]. Furthermore, EASSs sporicidal strong activity has been reported [18]. Additionally, EASSs have shown a priming effect on immune plant mechanisms, triggering a stronger activation of defense genes at each application. In addition, a biostimulant effect has been reported, which improves growth and production in important crops such as tobacco (*Nicotiana tabacum*) and apple (*Malus domestica*) [19].

The aim of this research was to evaluate the in vitro inhibition effect of EASSs on important fungal pathogens of the common bean plant (*Phaseolus vulgaris*) and the effect of EASSs in the symptoms of the infection caused by *Colletotrichum lindemuthianum* in common bean plants.

To our knowledge, no biological tests have been performed to evaluate the performance of the agricultural application of EASSs to ease microbial infection symptoms in plants.

## 2. Materials and Methods

### 2.1. Electrochemically Activated Salt Solutions (EASSs) Production Conditions

EASSs was produced from sodium chloride solution with water in an electrochemical cell; its anodes were of titanium recovered with metallic oxides based on ruthenium and iridium, and its cathodes were only of titanium. EASSs was produced as described in Table 1.

The free chlorine concentration (FCC) obtained was 2000 to 4000 ppm, depending on electrolysis time. Reported main chlorine species were HClO, ClO^−^, and Cl^−^, also depending to pH. Finally, ECAS was acidified with carbonic acid to achieve the desired pH. Its redox potential (ORP) increased with the solution’s acidity, from 0.860 V at pH = 8.75 to 1.050 V (at pH = 7).

### 2.2. Biological Material

The pathogenic isolates and common bean seeds (*Phaseolus vulgaris*) “c.v. pinto” were kindly provided by the Instituto de Investigaciones Forestales, Agrícolas y Pecuarias (INIFAP) Campo experimental Bajío, Celaya Guanajuato, México. In vitro tests were carried out in the phytopathology and molecular biology laboratories of the Center for Applied Research in Biosystems (CARB-CIAB) of the Faculty of Engineering of the Autonomous University of Querétaro, Campus Amazcala, El Marqués, Mexico. The in vivo tests were conducted in a 600 m^2^ multi-tunnel greenhouse on the same Campus.

### 2.3. In Vitro Inhibition Test

A solution of potato dextrose agar was made following the manufacturer instructions, and, after autoclave-sterilization, EASS was added to final concentrations of 25, 50, 75, 150, 250, 500, and 1000 ppm of FCC for each treatment (Table 2). Various pathogens, such as *Colletotrichum lindemuthianum*, *Macrophomina phaseolina*, *Phytium* sp., *Sclerotium rolfsii*, *Rhizoctonia solani*, and *Fusarium oxysporum*, were used in the tests to assess EASS in *Colletrotichum lindemuthiamun.* Next, 10 plates per concentration were inoculated with each pathogen by transferring a piece of an earlier established culture and placing it in the center of the sterile plate. In addition, a positive control without treatment and negative control samples, treated with a commercial fungicidal such as (CERCOBIN^®^ 50SC de CERTIS 15 g/L), was made (Table 2).

A final number of 540 plates was established, and all were incubated at room temperature for 4 weeks or until the mycelium had covered the plate (Final measured). Each week, the diameter of the colony in each plate was measured with a Vernier caliper.

### 2.4. Pathogenicity Test

*Colletotrichum lindemuthianum* was selected for further in vivo bioassays because of its commercial importance and high rate of EASSs-mediated inhibition. Seeds of common bean were germinated and, when they showed 2–3 true leaves, plantlets were transplanted into polyethylene bags with commercial substrate (peat moss and tezontle) and grown under greenhouse conditions. After 1 week, the plants were inoculated with *Colletotrichum lindemuthianum* by foliar spraying a solution with 1.2 × 10^10^ UFC of the pathogen. The EASSs treatments (Table 3) were applied at different concentrations (12.5, 25, 50, and 100 ppm of FCC); additionally, a group of plants was treated only with a commercial pesticide (CERCOBIN^®^ 50SC de CERTIS) (as a negative control) and another group of plants was inoculated with the pathogen without treatment (as a positive control). The six different groups were treated by spraying each solution on all the leaves of the plant until the drop point.

In total, 4 blocks of 15 plants underwent each treatment. Plants were maintained for 30 days in a greenhouse with controlled humidity and temperature values of 40% ± 10 and 33 °C ± 10 during the day and 70% ± 10 and 10 °C ± 10 during the night; they were irrigated with a standard Steiner solution at 30% concentration.

### 2.5. Plant Morphological Variables and Severity Level

The plants were monitored as growers, and the data collection was carried out manually every eight days, with a digital Vernier to measure the stem thickness and a meter to measure the height of the plant.

To evaluate the severity of the infection by *Colletotrichum lindemuthianum* in common bean plants, a severity scale was established, and the symptoms of the plants were rated based on a scale adapted from [20].

Each plant obtained a severity level from 0 to 5, where 0 described vigorous and healthy plants with no visible symptoms and 5 described plants with visible black spots of at least half the size of the leaves.

### 2.6. Plant Enzymatic and Antioxidant Variables

For each sample, 5 random leaves per treatment were collected and stored at −80 °C until their analysis. Once in the laboratory, an enzymatic extract was obtained as follows: the vegetal samples were homogenized in a mortar with a pestle and liquid nitrogen; once a fine powder was obtained, 0.3 g was measured and mixed with 1 mL of extraction buffer (phosphate buffer, 7.8 pH); the mixture then was vortexed for 2 min and centrifuged for 15 min at 12,000 rpm and 4 °C in a microcentrifuge. The resulting supernatant was stored at 4 °C as the enzymatic extract (EE).

The enzymatic and antioxidant activities of the samples were determined by spectroscopy through specific reactions in a spectrophotometer multiskan SkyHigh from Thermo Scientific (Waltham, MA, USA). First, the phenylalanine ammonia lyase enzymatic activity (PAL) was measured, as reported by [21], with modifications by the presence of cinnamic acid in the sample as a result of the L-phenylalaine catalysis by this enzyme. Then, in one well of a 96-well plate, 20 µL of EE were mixed with 230 µL of reaction buffer (0.1 M borate, 10 mM L-Phenylalanine, pH 8.8) and incubated at 40 °C for 60 min. Then, 50 µL of chlorhydric acid (HCl) was added to stop the reaction, which was set for 10 min at room temperature. The absorbance was read at 290 nm.

To determine the catalase activity (CAT), the catalysis of hydrogen peroxide was monitored over time as follows: 200 µL of reaction buffer (50 mM potassium phosphate, pH 8.0), 20 µL of hydrogen peroxide, and 10 µL of EE were mixed in a well of a 96-well plate and immediately placed to start the absorbance reading at 240 nm, once per minute for 6 min. This methodology was based on what was reported by [22]. The enzymatic activity was calculated with Formula (1):U = [(∆A)(Vt)] ÷ [(Ɛ)(Ve)](1)
where ∆A is the change in absorbance per minute, Vt is the total volume, Ɛ is the extinction coefficient of hydrogen peroxide, and Ve is the added EE.

Later, the enzymatic activity of superoxide dismutase (SOD) was measured indirectly by determining the amount of inhibition of nitroblue tetrazolium (NBT) by the superoxide ion. In a glass assay tube, 1.5 mL of reaction buffer (0.05 M potassium phosphate, pH 7.8) was mixed with 0.3 mL of 0.1 mM EDTA-Na, 0.3 mL of 0.13 M methionine, 0.3 mL of 0.75 mM of NBT, 0.3 mL of 0.02 mM riboflavin, 0.05 mL of EE, and 0.25 mL of distilled water. The tubes were mixed by inversion and exposed to fluorescent light for 30 min. After this, 250 µL of each mixture was placed in a well of a 96-well plate, and the absorbance was read at 560 nm, as reported by [23].

Furthermore, the radical scavenging activity (RSA) was determined by the DDPH method developed in [24]. First, 0.5 mL of EE and 0.5 mL of 0.1 mM DPPH were added to methanol to yield a final volume of 1.5 mL and were vortexed for 2 min. Then, the mixture was incubated for 30 min at room temperature while protected from light. When the time was up, 250 µL of the mix was placed in a well of a 96-well plate and absorbance was read at 525 nm. The following formula (Formula (2)) was used to calculate the percentage of antioxidants or RSA:% of antioxidant activity = [(AC − AT) ÷ AC] × 100(2)
where AC is the absorbance of the control and AT is the absorbance of treatment.

The Trolox equivalent antioxidant capacity (TEAC) activity was determined by obtaining a solution of ABTS radicals through mixing a 7 mM ABTS solution with 2.45 nM potassium persulfate and phosphate buffer (pH 7.4) until an absorbance of 0.35–0.4 at 734 nm was reached. Once the radical solution was obtained, 3 mL of it was mixed with 150 µL of EE and the absorbance was read each minute until 6 measurements were made at 734 nm. This methodology was based on what was reported by [25].

Finally, the content of proline was measured in leaves similarly to the procedure reported by [26], by mixing 1 mL of EE with 1 mL of glacial acetic acid and 1 mL of 0.5% ninhydrin reagent. After vortexing, the mixture was boiled for 30 min and then cooled. The mixture was phase-separated by the addition of 3 mL of toluene, and 250 µL of the upper phase was collected, placed in a well of a 96-well plate, and read on a spectrophotometer at 520 nm. The amount of proline was calculated by measuring the absorbance of a proline standard curve.

### 2.7. Statistical Analysis

For each parametric variable, a one-way ANOVA was performed to statistically confirm differences between treatments and, for severity, a Wilcoxon test was performed. Tukey’s test (α = 0.05) was used to identify different groups. For statistical analyses, the software JMP version 13.2.0 (JMP statistical discovery Cary, NC, USA) was used.

## 3. Results

### 3.1. In Vitro Inhibition Test

As observed in Figure 1, *Colletorichum lindemuthianum* inhibited development by the presence of Electrochemically Activated Salt Solutions (EASSs) at 1000 ppm of free chlorine concentration (FCC), which was equal to the negative control (chemical fungicide), from the first week of incubation until the end of the test. The culture was assessed weekly until the pathogen entirely covered the plates. *Colletotrichum lindemuthianum* grew in EASSs treatments from 25 to 500 ppm FCC, representing resistance, but even in the third week evaluated, none of the treated plaques developed similarly to the untreated control.

As shown in the general results of the experiment, the was an effect of the EASSs on the other pathogens in vitro. The most vulnerable pathogens were *Macrophomina phasedine* and *Phytium* sp., where their mycelia showed the fastest growth, leaving the petri dish covered from the first week of evaluation; the EASSs at 500 and 1000 ppm FCC were the ones with the lowest growth, which was even similar to the negative control.

In *Sclerotium rolfsii* and *Rhizoctonia solani*, the positive control covered the petri dish in the second week, and the inhibition was greater in the first week of the assay, decreasing over time; the concentration of EASSs of 1000 ppm of FCC was the one that remained inhibitory, along with the negative control. Finally, for *F. oxysporum,* the mycelia covered the cane in the second week; this grew in all EASS treatments, but showed greater inhibition than the positive control until the second and last week (Table 4).

### 3.2. Plant Morphological Variables

As described before, the application of EASSs to in vitro cultures of the phytopathogen fungus *Colletotrichum lindemuthianum* was the most effective treatment. The next step was to evaluate the performance of this treatment over an in vivo assay with common bean plants infected with an isolate of *Colletotrichum lindemuthianum*. The EASSs were applied in lower concentrations than tested before due to their proven efficiency. Morphological variables were measured to compare the plant growth through the treatments; Figure 2 represents the statistical results at 30 days of bean culture with infection by anthracnose. In these cases, no statistical differences were observed between the positive control (infected), the negative control (infected with a chemical fungicide), and EASSs 12.5 and 25 ppm. Furthermore, there was a significant difference in plant growth on EASSs 50 ppm and stem thickness on EASSs 100 ppm.

### 3.3. Plant Severity Levels

As expected, the symptoms of the plants in the pathogenicity test were the most affected in the positive control (inoculated plants with no treatments) (Figure 3). Moreover, lower symptoms were observed in the inoculated plants treated with a chemical fungicide. Regarding the EASSs-treated groups, at the first time of evaluation, only the higher concentration (100 ppm) showed differences from the negative control, meaning that the plants treated with the other concentrations showed the same levels of severity as the infected plants treated with a chemical fungicide. On the other hand, at the second time of evaluation, the highest and second-highest concentrations showed the same severity levels as the positive control.

### 3.4. Plant Enzymatic and Antioxidant Variables

The enzymatic activity of phenylalanine ammonia lyase (PAL) measured 15 days after the application of the treatments was higher in the plants treated with EASSs at 12.5 and 100 ppm and inoculated with *Colletotrichum lindemuthianum*. The following concentration (25 ppm) showed the lowest activity in the same statistical group as the plants treated with EASSs plants inoculated with the pathogen and without treatment (positive control) (Figure 4). The other concentrations of EASSs treatments remained between the lower concentration and the positive control for the same length of time as the inoculated plants treated with a chemical product (negative control). The results of this enzyme 30 days after EASSs treatments show that the highest concentration of PAL was acquired in EASSs 12.5, 24, and 50 ppm, The lowest PAL activity was obtained in the positive control and the treatments. This response showed an apparent relationship indirectly with the concentration, showing, at a higher concentration, less activity (Figure 4A).

The other measured enzymatic activity was from the enzyme catalase (CAT), which showed differences between the group treated with EASSs at 12.5 ppm and without treatment (control positive) at 15 days after the treatments, as shown in Figure 4B. The group treated with EASSs at 12.5 ppm was the lowest CAT activity. At 30 days, no statistical differences were obtained between groups.

The presence of proline in the leaves was determined, showing an increased concentration along the measured days.

The proline concentration at 30 days ranged from 94.5 to 142.5 ug/mg protein; at 15 days, the range of concentrations was 23.9 to 31 ug/mg protein. At 15 days, the treatments with EASSs showed more proline than the two controls. Contrastingly, at 30 days after treatments, the EASSs-treated plants showed lower proline content than positive control and lower than or the same as negative control. For this determination, the group treated with EASSz at 100 ppm behaved differently, showing the highest proline content at 15 days but the lowest proline content at 30 days (Figure 4C).

Finally, in this work, the non-enzymatic antioxidant activity was also measured by determining ABTS and DDPH presence; both have the same comportment (Figure 4D,E). For these variables, we found similar results for EASSs at 25 ppm, with the highest activity at 15 days and still being in the group with the highest activities at 30 days. The groups with the lower concentrations of EASSs presented the higher antioxidant activities of ABTS and DPPH, even higher than both of controls for the two measured times.

## 4. Discussion

Currently, the search for safer alternatives to promote agricultural yield has been increasing. The common bean is a highly diverse crop that adapts to very different climates, resulting in multiple available varieties. This diversity also suggests high diversity in the defense mechanisms of each variety, which are available to activation with the correct elicitation design. In this paper, we propose using EASSs as plants elicitors by the activation of multiple biosynthetic pathways, resulting in defense and secondary metabolite activation, both highly desired in agricultural production. Furthermore, this alternative is not toxic for the field workers, environment, or the final consumer. Here, we evaluated the in vitro effect of an EASSs directly applied to several phytopathogenic microbes, and *Colletotrichum lindemuthianum* was selected for further in vivo evaluations of EASSs-elicited plant–pathogen interaction, bringing this work a step further into the application of EASSs as an agricultural treatment in primary production systems.

As expected, the best EASSs concentrations for microbe inhibition were 500 and 1000 ppm, but all the concentrations showed some inhibition. The behavior of inhibition depended on the evaluated microorganism, but the stronger inhibition generally resulted at the beginning of the test. This could be influenced by a lower persistence of the EASSs in the environment, which turned out to be an advantage to become a safer agricultural product, but this suggests the need to apply it more than once in the crop cycle.

The morphological effect in plants was evaluated, with no differences reported. This was not expected, as the activation of biological pathways requires energy investment; commonly, this energy is taken from biological events such as plant growth and development. With this result, we suggest that the EASSs elicitation mechanism allows the plants to efficiently distribute the energy in order to continue normal growth and development, while at the same time to activate defense-signaling cascades. This should not lower the yield of EASSs-treated bean crops. In this research on the common bean “c.v. pinto”, our results indicated that the most significant disease symptoms occurred in the plants without treatment, which represents the susceptibility to the disease. This can refer to the permanence of the seed and the pathogenic spread to other regions and crop beans [8,9,10,27].

Interestingly, the infection with *Colletotrichum lindemuthianum* showed higher severity in plants treated with the most concentrated EASSs; this suggests important levels of distress in plants due to the EASSs treatment in addition to the pathogenic infection. Further, the lower concentrations of EASSs (12.5, 25, and 50 ppm) showed statistically identical results in the severity of plants. These results provide information related to the hormetic behavior of the elicitation effect of EASSs in common bean plants, which, at the same time, allows the researchers to design controlled elicitation plans in future agricultural treatments.

The phenylpropanoid pathway (ppp) and antioxidant activity are highly desired variables for both crop producers and final consumers, especially in common bean, where important levels have been reported. Researchers have successfully elicited these two biological systems in common bean by ultrasound [28], thermal variations [29], and NaCl and glutaminc acid application [30]. Now, the present work allowed us to confirm an elicitation of ppp and antioxidant activity by EASSs foliar treatment, which remain active even 30 days after the application of the treatment.

The results showed statistically higher antioxidant activity for ABTS and DPPH but not for CAT, which may suggest that EASSs elicits the activation of a non-enzymatic antioxidant system that remains significant until 30 days after the application of the treatment. Therefore, the activation of synthetic pathways such as ppp—where PAL is one of the earliest involved enzymes to raise the concentration of non-enzymatic antioxidants—may be the main defense mechanism activated by the application of EASS in common bean [31]. Supporting this possibility, we can also observe an accumulation of proline in some EASSs treatments; proline is an amino acid with several important roles in plant metabolism, one of them being service as an antioxidative defense molecule [32].

By these results, we can propose that the main reason for a difference in the severity of the infection is the elicitation of the plants and activation of defenses, rather than the antimicrobial activity that EASSs have. However, as the elicitation of plants may affect a specific kind of pathogen, it would be interesting to test different pathogenic microorganisms to validate the EASSs efficiency as a treatment for pest management.

In this work, the effect of EASSs on phytopathogen survival and infection severity was evaluated. The results suggest that EASSs applied at low concentrations are viable, functional, organic treatments that help to manage fungal pests in crops.

## 5. Conclusions

In this work, we were able to detect the activation of multiple defense plant mechanisms, such as the activity of the enzyme phenylalanine ammonia lyase, the antioxidant system, and the production of proline.

Our data suggest the activated defenses were strong enough to successfully decrease the severity of the disease at the lower concentrations of the treatment.

Applying EASSs in common bean crops could be an effective treatment for *Colletotrichum* infection management. However, treatment doses should be carefully selected and supplied to avoid distress in the plants.

## Figures and Tables

**Figure 1 biology-12-00964-f001:**
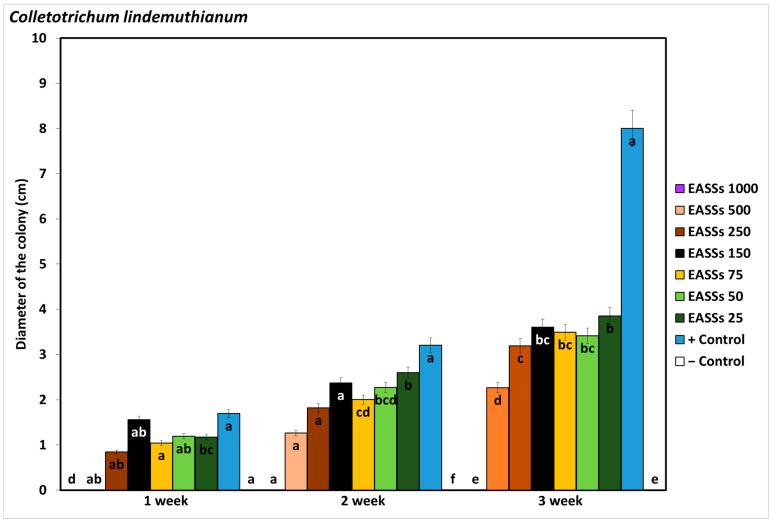
Effect of EASSs in vitro against *Colletotrichum lindemuthianum*. Different letters indicate significant differences according to Tukey´s test (α = 0.05).

**Figure 2 biology-12-00964-f002:**
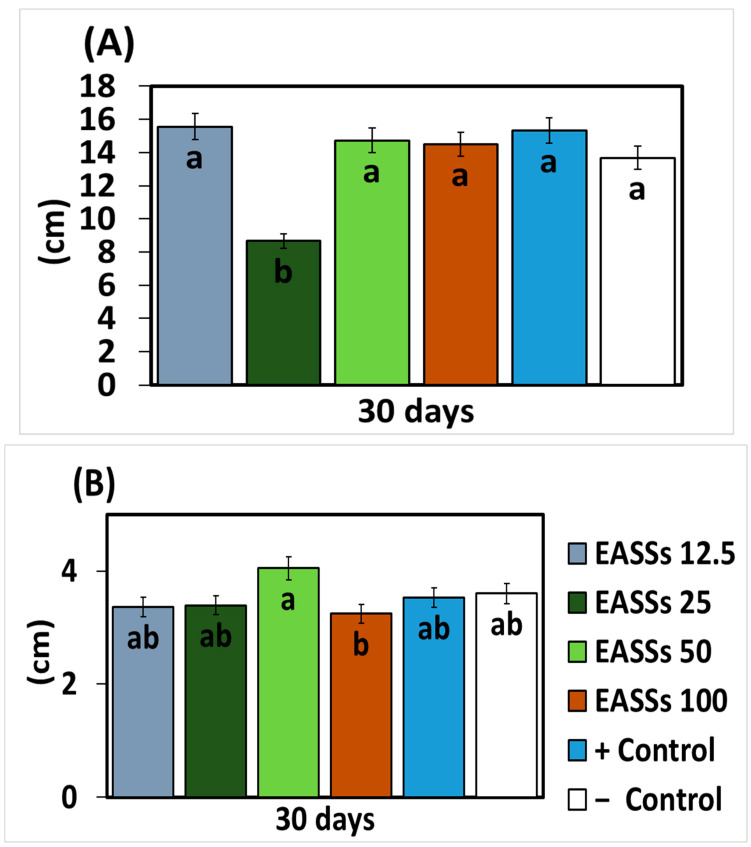
Effect of EASSs treatments on morphological variables ((**A**) Plant height and (**B**) Stem thickness) of bean plants inoculated with *Colletotrichum lindemuthianum* at 30 days. Different letters indicate significant differences according to Tukey’s test (α = 0.05).

**Figure 3 biology-12-00964-f003:**
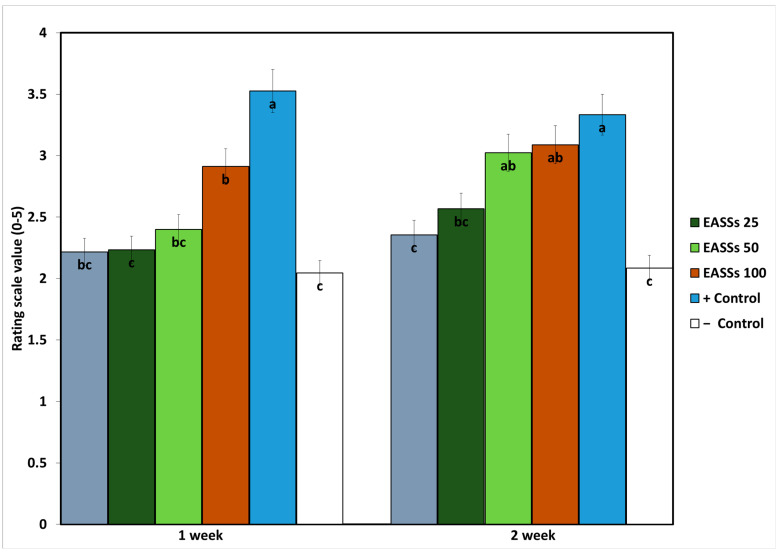
Severity progression of *Colletotrichum lindemuthianum* on bean growth treated with EASSs. Different letters indicate significant differences according to Tukey’s test (α = 0.05).

**Figure 4 biology-12-00964-f004:**
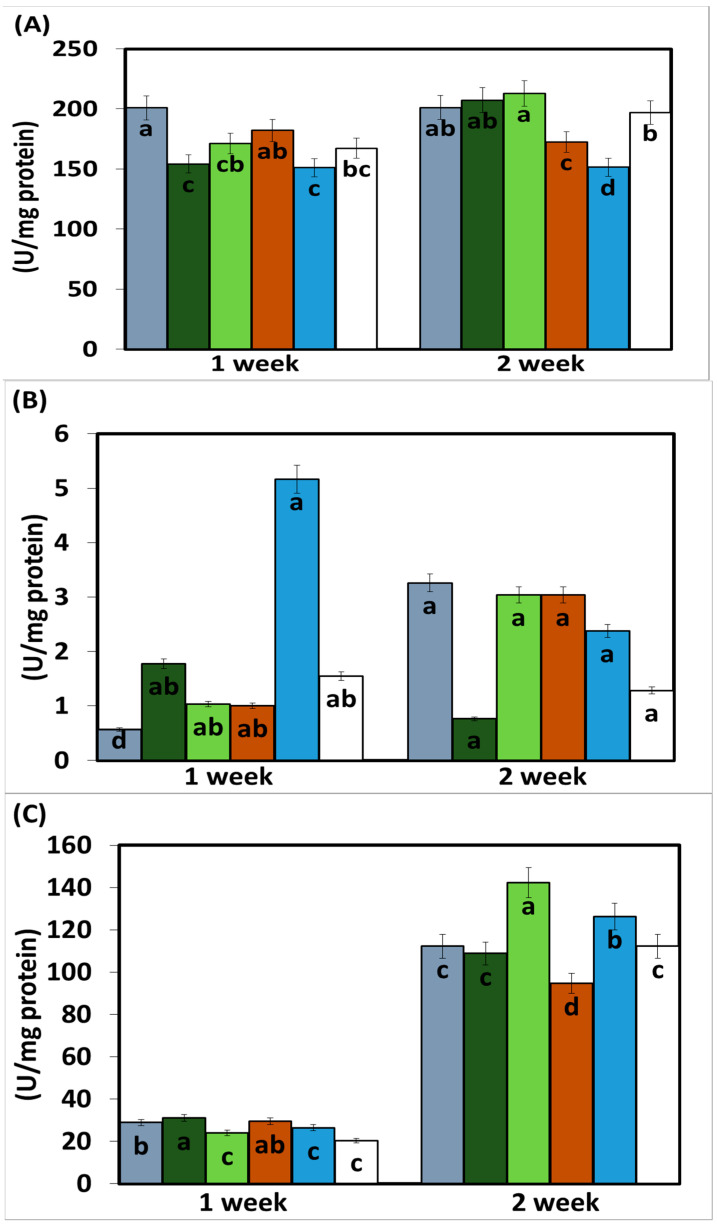
Enzymatic and antioxidant activity ((**A**) PAL, (**B**) CAT, (**C**) Proline, (**D**) ABTS and (**E**) DDPH) on bean infected with *Colletotrichum lindemuthianum* and treated with EASSs. Different letters indicate significant differences according to Tukey’s test (α = 0.05).

**Table 1 biology-12-00964-t001:** EASSs production and operation variables at 24 h.

Operation Variables	Quantities
Sodium chloride concentration	8 g/L
Water volume	250 L
Initial free chloride concentration	0 mg/L
Final free chloride concentration	3592 mg/L
Electrolysis time	24 h
Amperage	40 A
Voltage	4.88 V
Initial temperature	20 °C
Final temperature	22 °C
Oxide-reduction potential (ORP)	871 mV

**Table 2 biology-12-00964-t002:** Treatments applied during in vitro tests.

Treatment	Description
Negative control	CERCOBIN^®^ 50SC de CERTIS 0.15 g/L
Positive control	Distilled water
EASSs 25	EASSs 25 ppm
EASSs 50	EASSs 50 ppm
EASSs 75	EASSs 75 ppm
EASSs 150	EASSs 150 ppm
EASSs 250	EASSs 250 ppm
EASSs 500	EASSs 500 ppm
EASSs 1000	EASSs 1000 ppm

**Table 3 biology-12-00964-t003:** Treatments applied on in vivo bioassays.

Treatment	Description
Negative control	CERCOBIN^®^ 50SC de CERTIS 0.15 g/L
Positive control	Distilled water
EASSs 12.5	EASSs 12.5 ppm
EASSs 25	EASSs 25 ppm
EASSs 50	EASSs 50 ppm
EASSs 100	EASSs 100 ppm

**Table 4 biology-12-00964-t004:** Effect of EASSs in vitro against multiple pathogens. Different letters indicate significant differences according to Tukey’s test (α = 0.05).

Treatments	Pathogens				
	*M. phasedine*	*Phytium* sp.	*S. rolfsii*	*R. solani*	*F. oxysporum*
EASSs 1000	0.568 cd	0.825 b	0 d	1.280 c	4.900 c
EASSs 500	1.590 c	0.705 b	3.400 c	7.173 b	4.893 c
EASSs 250	3.448 b	7.243 a	6.455 ab	8 a	5.678 bc
EASSs 150	7.750 a	5.970 a	7.290 ab	8 a	6.215 b
EASSs 75	6.798 a	7.605 a	6.853 ab	8 a	6.225 b
EASSs 50	7.167 a	7.945 a	5.995 b	8 a	6.273 b
EASSs 25	7.165 a	8 a	7.463 ab	8 a	5.825 bc
+Control	7.807 a	7.730 a	8 a	8 a	7.880 a
−Control	0 d	2 c	0 d	0 d	0 d

## Data Availability

Not applicable.

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
