# Peer review of "Management of Antracnosis with Electrochemically Activated Salt Solutions (EASSs) on Bean Culture"

_biology, 2023, doi:10.3390/biology12070964_

Round 1
Reviewer 1 Report
Dear authors,
your scientific article is very interesting and raises an important issue. I really appreciate the innovative nature of the work and the contribution that has been made to the research results.
However, I have some suggestions for the final version of the work.
I would ask you to specify in the Methods Chapter which variety of bean was used in your research. I consider this very important, especially since you started the discussion in the chapter with the paragraph "Currently, the search for safer alternatives to promote agricultural yield has been increasing.
The common bean is a highly diverse crop that adapts to very different weathers resulting in multiple available varieties.This diversity also suggests high diversity in the defense mechanisms of each variety, available to activate with the correct elicitation design".
So it would be worth characterizing the variety you used in your research with justification why me, referring to its potential susceptibility to diseases and possible economic importance (is this variety very willingly cultivated by farmers?).
I found no linguistic or stylistic errors in English. The work is well written and the sentences are very clearly formulated.
Author Response
I found no linguistic or stylistic errors in English. The work is well written and the sentences are very clearly formulated.
Point 1: I would ask you to specify in the Methods Chapter which variety of bean was used in your research.
Response 1:
Thanks for your comments,
In Methods Chapter section 2.2. Biological material.
In the first sentence said:
“common bean seeds (Phaseolus vulgaris) “Pinto” were kindly provided by the Instituto de Investigaciones Forestales, Agrícolas y Pecuarias (INIFAP) Campo experimental Bajío, Celaya Guanajuato México.
and was added to:
“common bean seeds (Phaseolus vulgaris) “c.v. Pinto” were kindly provided by…”.
Point 2: I consider this very important, especially since you started the discussion in the chapter with the paragraph "Currently, the search for safer alternatives to promote agricultural yield has been increasing.
The common bean is a highly diverse crop that adapts to very different weathers resulting in multiple available varieties. This diversity also suggests high diversity in the defense mechanisms of each variety, available to activate with the correct elicitation design.
Response 2:
In Line 334 to 337, as we added a paragraph and references:
“In this research on the common bean “ c.v. pinto”, our results indicated that the most significant disease symptoms occurred in the plant without treatment, which represents the susceptibility to the disease. It can mean the permanence of the seed and the pathogenic spread to other regions and crop beans [8,9,10,27].”
Point 3: So it would be worth characterizing the variety you used in your research with justification why me, referring to its potential susceptibility to diseases and possible economic importance (is this variety very willingly cultivated by farmers?).
Response 3: This information is important to make it known that the anthracnose disease continues to cause losses in bean producers, and the bean variety selected for this investigation resulted in susceptibility to the disease.

Reviewer 2 Report
In this manuscript, the authors studied the effect of electrochemically activated salt solutions (EA) on anthracnose in bean cultures.They demonstrated that the presence of EASS in the growth media at the beginning of the experiment had a strong inhibitory effect on some of the fungi studied.
They concluded that EASS is a promising way to control the disease in a more economical and environmentally friendly way.
My comments and suggestions can be found in the study itself.
In general, the topic of the paper is interesting and potentially useful. Although the paper thematically meets the requirements of the journal, some additional analysis should be done to fully address the conclusion of the paper.
In general, the results/figures presented are of poor quality and not clearly presented, especially in Fig. 1 where it is not understandable what the values of EASS100, EASS500 and -control are.
Results on Macrophomina phasedine, Phytium sp., Sclerotium rolfsii and Rhizoctonia solani are missing. The description of Figure 2. is not complete and correct.
In addition, it would be useful to evaluate the production of some secondary metabolites, as the paper itself states.
"In this work, we propose to use EASS as plant elicitors by activating multiple biosynthetic pathways, leading to the activation of defenses and secondary metabolites, both of which are highly desirable in agricultural production."
It would be good to evaluate hydrogen peroxide, MDA and other antioxidant enzymes (APX. PPX, SOD). It would be good to analyze the elements in the bean to see if they change with treatment, etc.
In my opinion, the potential of this work has not been fully realized, and a comprehensive study is needed to refine the work.

Editing of English language is required
Author Response
My comments and suggestions can be found in the study itself.
In general, the topic of the paper is interesting and potentially useful. Although the paper thematically meets the requirements of the journal, some additional analysis should be done to fully address the conclusion of the paper.
Point 1: In general, the results/figures presented are of poor quality and not clearly presented, especially in Fig. 1 where it is not understandable what the values of EASS1000, EASS 500 and -control are.
Response 1:
In Methodology Section 2.2; Line 114-116; the original text indicates the values of treatments :
“A solution of potato dextrose agar was made following the manufacturer instructions and after autoclave-sterilization, EASS were added to final concentrations of 25, 50, 75, 150, 250, 500 and 1000 ppm of FCC respectively for each treatment. ”
And was added a table 2, to describe the values of figure 1:
Table 2. Treatments applied on in vitro test.
|
Treatment |
Description |
|
Negative control |
CERCOBIN® 50SC de CERTIS 0.15g/L |
|
Positive control |
Distilled water |
|
EASS 25 |
EASS 25 ppm |
|
EASS 50 |
EASS 50 ppm |
|
EASS 75 |
EASS 75 ppm |
|
EASS 150 |
EASS 150 ppm |
|
EASS 250 |
EASS 250 ppm |
|
EASS 500 |
EASS 500 ppm |
|
EASS 1000 |
EASS 1000 ppm |
And all changes in figures were carried out.
Point 2: Results on Macrophomina phasedine, Phytium sp., Sclerotium rolfsii and Rhizoctonia solani are missing.
Response 2:
In Results Section 3.1, in line 236-246 saids:
“As general results of the experiment of the other pathogens in vitro were obtained; The most vulnerable pathogens were Macrophomina phasedine and Phytium sp., where their mycelium showed the fastest growth, leaving the petri dish covered from the first week of evaluation, being the EASS at 500 and 1000 ppm FCC the ones with the lowest growth, even similar to the negative control.
In Sclerotium rolfsii and Rhizoctonia solani, the positive control covered the petri dish in the second week, and the inhibition was greater in the first week of the assay, decreasing over time, being the concentrations of EASS 1000 ppm of FCC the one that remained inhibitory. together with the negative control. And finally, F. oxysporum, the mycelium covered the cane in the second week; grew in all EASS treatments but showed greater inhibition than the positive control until the second and last week.”
Point 3: The description of Figure 2. is not complete and correct.
Response 3: We changed the description of Figure 2 to:
“Figure 2. Effect of EASS treatments on morphological variables on bean plants inoculated with Colletotrichum lindemuthianum at 30 days . Different letters indicate significant differences according to Tukey´s test (α =0.05).”
We changed the description and hope it varies according to your request.
Point 4: In addition, it would be useful to evaluate the production of some secondary metabolites, as the paper itself states.
"In this work, we propose to use EASS as plant elicitors by activating multiple biosynthetic pathways, leading to the activation of defenses and secondary metabolites, both of which are highly desirable in agricultural production."
It would be good to evaluate hydrogen peroxide, MDA and other antioxidant enzymes (APX. PPX, SOD). It would be good to analyze the elements in the bean to see if they change with treatment, etc.
In my opinion, the potential of this work has not been fully realized, and a comprehensive study is needed to refine the work.
Respond 4. At the time, it would be essential to add the analyzes that you suggest, and we know it can complement the research and should be with future work; however, at this moment, we do not have the necessary resources to carry them out, but we think that it is needed to know this information does not exist and can use for future research.

Reviewer 3 Report
The whole manuscript needs to be revised, there are various punctuation errors and some sentences are not clear.
-line 36: "on plants such as tobacco and apple"; it is a repetition of the summary, please eliminate it.
-line 50: some keywords are present in the title, please change them
Material & Methods:
-section 2.2: this section is very poor, please add some more information or add this part to the text of 2.3 and 2.4.
-line 121: please specify the number of negative controls or eliminate "several".
-line 133: move here the information about the green-house conditions.
-section 2.5: This section is very poor, please add more information or combine it with the section 2.7 and modify the title.
-line 147: which several variables? Please specify or delete it
-line 163: eliminate "for" or "during"
Results:
Figure 1: please specify in the legend that the negative control contains only the fungicide. Add in the caption of each figure the description of the treatments
-line 229: restructure the sentence
-line 229-229: please show the results in a graph or in a table
-Figure 3: the legend is not complete
-Figure 4: put the graphs in the same order of the text or change the text
Discussion:
-lines 324-326: please delete this sentence, it is too speculative
-please add some more information about the role of the single analyzed enzymatic activities in the plant metabolism/plant defence mechanisms
The whole manuscript should be revised, there are various punctuation errors and some sentences are not clear.
Author Response
The whole manuscript needs to be revised, there are various punctuation errors and some sentences are not clear.
Thank you for your comments. We will take them into account in future publications and show the changes made in this article:
Point 1: -line 36: "on plants such as tobacco and apple"; it is a repetition of the summary, please eliminate it.
Response 1: We eliminated it, and thanks for your comments.
Point 2: -line 50: some keywords are present in the title, please change them
Response 2: The change was carried out
Point 3: Material & Methods:
-section 2.2: this section is very poor, please add some more information or add this part to the text of 2.3 and 2.4.
Response 3: We added the next paragraph:
“In vitro tests were carried out in the phytopathology and molecular biology laboratories of the Center for Applied Research in Biosystems (CARB-CIAB) of the Faculty of Engineering of the Autonomous University of Querétaro, Campus Amazcala, El Marqués, Mexico. In vivo tests were conducted in a 600 m2 multi-tunnel greenhouse with aspersion of irrigation, on the same Campus.”
Point 4: -line 121: please specify the number of negative controls or eliminate "several".
Response 4: We eliminated several.
Point 5: -line 133: move here the information about the green-house conditions
Response 5: We put the greenhouse information on response three and eliminated that part for the line.
Point 6: -section 2.5: This section is very poor, please add more information or combine it with the section 2.7 and modify the title.
Response 6: We combined it with section 2.6. and change the order of the sections.
Point 7: -line 163: eliminate "for" or "during"
Response 7: We eliminated the word “for“.
Point 8: Results:
Figure 1: please specify in the legend that the negative control contains only the fungicide. Add in the caption of each figure the description of the treatments
Response 8: We added a table 2 that specifies the meanings of the controls and treatments in the methodology section 2.3 and hopefully was according to your comments.
And below is Table 2, which describes the values of Figure 1:
Table 2. Treatments applied on in vitro test.
|
Treatment |
Description |
|
Negative control |
CERCOBIN® 50SC de CERTIS 0.15g/L |
|
Positive control |
Distilled water |
|
EASS 25 |
EASS 25 ppm |
|
EASS 50 |
EASS 50 ppm |
|
EASS 75 |
EASS 75 ppm |
|
EASS 150 |
EASS 150 ppm |
|
EASS 250 |
EASS 250 ppm |
|
EASS 500 |
EASS 500 ppm |
|
EASS 1000 |
EASS 1000 ppm |
Point 9: -line 229: restructure the sentence
Response 9: We changed the sentence to:
“As shown in the general results of the experiment, the effect of the EASS on the other pathogens in vitro;”
Point 10: -line 229-229: please show the results in a graph or in a table
Response 10: Thank you for your comments, we decided to show the results this way because this investigation’s pathogen is Colletotrichum, but we considered it essential to describe the effects of the other pathogens in vitro. All the figures and tables are of the main pathogen. We know this information does not exist, and this description may be helpful for further research.
Point 11: - Figure 3: the legend is not complete
Response 11: We changed the legend of figure 3:
“Figure 3. Severity progression of Colletotrichum lindemuthianum on bean growth treated with EASS.”
We changed the description and hope it varies according to your request.
Point 12: -Figure 4: put the graphs in the same order of the text or change the text
Response 12: We changed the graphs and reordered the text.
Point 13: Discussion:
-lines 324-326: please delete this sentence, it is too speculative
Response 13: We change the sentence meaning by avoiding generalizing; apologies, and we specify that this happened in the infected bean plants.
Modified text::
“With this result, the EASS elicitation mechanism allows bean plants to efficiently distribute energy against phytopathogens to continue their average growth and development while activating defense signaling cascades.”
Point 14: -please add some more information about the role of the single analyzed enzymatic activities in the plant metabolism/plant defence mechanisms
Response 14: We consider showing the information in lines 335 to 350 in the original version.
“The phenylpropanoid pathway (ppp) and antioxidant activity are highly desired variables for both crop producers and final consumers, especially in common bean where important levels have been reported. Researchers have successfully elicitated these two biological systems in common bean by ultrasound [27], thermal variations [28] and NaCl and Glutaminc acid application [29]. Now, the present work allowed us to confirm an elicitation of ppp and antioxidant activity by EASS foliar treatment which remain active even 30 days before the application of the treatment.
The results showed statistically higher antioxidant activity for ABTS and DPPH but not for CAT, this may suggest that EASS elicitate the activation of a non-enzymatic antioxidant system that remained significant until 30 days after the application of the treatment. Therefore, the activation of synthetic pathways such a ppp where PAL is one of the earliest involved enzymes that raises the concentration of non-enzymatic antioxidants may be the main defense mechanism activated by the application of EASS in common bean [30]. Supporting this possibility, we can also observe accumulation of proline in some EASS treatments, the proline is an aminoacid with several important roles in plants metabolism, one of them as antioxidative defense molecule [31].”

Round 2
Reviewer 2 Report
Regarding the revised manuscript, I have a few points to address:
1. in vitro should be In italic through the text.
2. Regarding the results on Macrophomina phasedine, Phytium sp., Sclerotium rolfsii and Rhizoctonia solani, I can only read the text, but I do not see any figures with the results, and in the text the authors do not refer to any figure.
I recommend having the manuscript reviewed by a colleague who is fluent in English or using one of the editing services.
Author Response
Regarding the revised manuscript, I have a few points to address:
Point 1: in vitro should be In italic through the text.
Response 1: All changes were carried out.
Point 2: Regarding the results on Macrophomina phasedine, Phytium sp., Sclerotium rolfsii and Rhizoctonia solani, I can only read the text, but I do not see any figures with the results, and in the text the authors do not refer to any figure.
Response 2:
In Results Section 3.1, we added Table 4, which specifies the results of the multiple pathogens, according to the final measurements described, and hopefully was according to your comments.
And below is Table 4:
|
Table 4. Effect of EASS in vitro against multiple pathogens. Different letters indicate significant differences according to Tukey´s test (α =0.05). |
|||||
|
Treatments |
Pathogens
|
||||
|
|
M. phasedine |
Phytium sp. |
S. rolfsii |
R. solani |
F. oxysporum |
|
EASS 1000 |
0.568 cd |
0.825 b |
0 d |
1.280 c |
4.900 c |
|
EASS 500 |
1.590 c |
0.705 b |
3.400 c |
7.173 b |
4.893 c |
|
EASS 250 |
3.448 b |
7.243 a |
6.455 ab |
8 a |
5.678 bc |
|
EASS 150 |
7.750 a |
5.970 a |
7.290 ab |
8 a |
6.215 b |
|
EASS 75 |
6.798 a |
7.605 a |
6.853 ab |
8 a |
6.225 b |
|
EASS 50 |
7.167 a |
7.945 a |
5.995 b |
8 a |
6.273 b |
|
EASS 25 |
7.165 a |
8 a |
7.463 ab |
8 a |
5.825 bc |
|
+ Control |
7.807 a |
7.730 a |
8 a |
8 a |
7.880 a |
|
- Control |
0 d |
2 c |
0 d |
0 d |
0 d |
